# Glycolysis-Related SLC2A1 Is a Potential Pan-Cancer Biomarker for Prognosis and Immunotherapy

**DOI:** 10.3390/cancers14215344

**Published:** 2022-10-29

**Authors:** Haosheng Zheng, Guojie Long, Yuzhen Zheng, Xingping Yang, Weijie Cai, Shiyun He, Xianyu Qin, Hongying Liao

**Affiliations:** 1Department of Thoracic Surgery, Thoracic Cancer Center, The Sixth Affiliated Hospital, Sun Yat-sen University, Guangzhou 510655, China; 2Guangdong Research Institute of Gastroenterology, The Sixth Affiliated Hospital, Sun Yat-sen University, Guangzhou 510655, China; 3Department of Pancreatic Hepatobiliary Surgery, The Sixth Affiliated Hospital of Sun Yat-sen University, Guangzhou 510655, China

**Keywords:** SLC2A1, pan-cancer, glycometabolism, immune infiltration, biomarker, prognosis

## Abstract

**Simple Summary:**

Enhanced glycolysis is a major feature of cancer glycometabolism, and SLC2A1 is one of the pivotal genes in cancer glycometabolism. Although SLC2A1 plays an important role in the growth of many cancers, pan-cancer analysis allows us to more comprehensively and systematically understand the function and role of SLC2A1 in cancers. In this study, we found that SLC2A1 was highly expressed in most cancers, and resulted in poor prognosis. M6A methylation might be one of the important factors for the high expression level of SLC2A1. SLC2A1 not only enhanced cancer glycolysis, but also affected the tumor microenvironment. Notably, SLC2A1 was significantly and positively correlated with the T-cell-exhaustion biomarkers PD-L1 and CTLA4. Collectively, SLC2A1 may provide new strategies for pan-cancer treatment, especially cancer immunotherapy.

**Abstract:**

SLC2A1 plays a pivotal role in cancer glycometabolism. SLC2A1 has been proposed as a putative driver gene in various cancers. However, a pan-cancer analysis of SLC2A1 has not yet been performed. In this study, we explored the expression and prognosis of SLC2A1 in pan-cancer across multiple databases. We conducted genetic alteration, epigenetic, and functional enrichment analyses of SLC2A. We calculated the correlation between SLC2A1 and tumor microenvironment using the TCGA pan-cancer dataset. We observed high expression levels of SLC2A1 with poor prognosis in most cancers. The overall genetic alteration frequency of SLC2A1 was 1.8% in pan-cancer, and the SLC2A1 promoter was hypomethylation in several cancers. Most m6A-methylation-related genes positively correlated with the expression of SLC2A1 in 33 TCGA cancers. Moreover, SLC2A1 was mainly related to the functions including epithelial–mesenchymal transition, glycolysis, hypoxia, cell-cycle regulation, and DNA repair. Finally, SLC2A1 positively associated with neutrophils and cancer-associated fibroblasts in the tumor microenvironment of most cancers and significantly correlated with TMB and MSI in various cancers. Notably, SLC2A1 was remarkably positively correlated with PD-L1 and CTLA4 in most cancers. SLC2A1 might serve as an attractive pan-cancer biomarker for providing new insights into cancer therapeutics.

## 1. Introduction

Cancer is one of the leading causes of death in humans. Although much progress has been made in the treatment of cancer, the overall therapeutic effect is unsatisfactory. Newly diagnosed cases are also increasing, placing a huge burden on society [1]. In recent years, cancer immunotherapy has considerably progressed, providing a powerful tool for cancer treatment and improving the prognosis of cancer patients [2]. In 2017, the U.S. FDA approved pembrolizumab for solid tumors with high microsatellite instability or mismatch repair gene defects (MSI-H/dMMR). Pembrolizumab has also become the first antitumor immune drug based on pan-cancer biomarkers without paying attention to the cancer type [3]. Pan-cancer analysis can help us understand the commonalities among different cancer types and provide new ideas for the treatment of pan-cancer [4].

Glucose is one of the basic metabolites needed by animal and plant cells. Cancer cells require a large amount of energy from the body for malignant proliferation [5]. Aberrant energy metabolism is an important feature of cancer cells. Even with ample oxygen supply, most tumor cells prefer enhanced glycolysis instead of oxidative phosphorylation to produce ATP [6]. Metabolite reprogramming provides energy and biological materials, providing a growth advantage to tumor cells under hypoxia [7]. Therefore, cancer metabolic reprogramming is an important direction in the search for novel pan-cancer biomarkers

Solute carrier family 2 member 1 (SLC2A1) is known as glucose transporter 1 (GLUT1) [8]. SLC2A1 plays a crucial role in the process of cell glycometabolism, whether in cancer or normal cells [9]. SLC2A1 is highly expressed in many kinds of cancer, and the overexpression of SLC2A1 can promote the growth and metastasis of cancers, such as liver, lung, endometrial, oral, breast, and gastric cancers [10,11,12,13,14,15]. Although the overexpression of SLC2A1 can further enhance glycolysis and cell proliferation in various cancers, a comprehensive pan-cancer analysis on SLC2A1 is lacking.

In this study, data from public databases and our own data convincingly showed that the expression of SLC2A1 was significantly increased in pan-cancer and conferred a poor prognosis. We explored the potential mechanism of SLC2A1 in pan-cancer through bioinformatics analysis. We further examined the association between SLC2A1 and the immune cell infiltration score, immune checkpoints, TMB, and MSI. Our results comprehensively revealed the potential mechanism of SLC2A1 in pan-cancer, and they highlight the impact of SLC2A1 on the tumor microenvironment (TME) and cancer immunotherapy.

## 2. Materials and Methods

### 2.1. Data Collection

We downloaded transcriptome data and clinical information from the University of California Santa Cruz (UCSC) Xena browser (https://xena.ucsc.edu/, accessed on 14 July 2022) and the Genotype-Tissue Expression (GTEx) database (https://www.gtexportal.org/home/-index.html, accessed on 14 July 2022), which included 15,776 samples of 33 cancer types and normal tissues. The abbreviations of all cancer types are shown in Appendix A. Using the R package of “rma”, we transformed the whole data by log2(TPM +1), which we then filtered to remove missing and duplicated results. In addition, we searched 20 relative datasets from the Gene Expression Omnibus (GEO) database (https://www.ncbi.nlm.nih.gov/geo/, accessed on 14 July 2022) for validation. These datasets were GSE2088, GSE13507, GSE10927, GSE39001, GSE26566, GSE18520, GSE53757, GSE62452, GSE87211, GSE15605, GSE33630, GSE3218, GSE17025, GSE47861, GSE68468, GSE53625, GSE13601, GSE57927, GSE75037, and GSE26899. Detailed information on the GEO datasets is shown in Appendix A.

We collected 90 pairs of samples (30 LUAD, 30 ESCA, and 30 COAD) from the Sixth Affiliated Hospital of Sun Yat-Sen University. Each sample contained paired tumors and adjacent normal tissues. The study was approved by the Ethics Committees of the Sixth Affiliated Hospital of Sun Yat-Sen University.

Finally, we used RT-qPCR method to validate the differential expression of SLC2A1 in LUAD and ESCA between cancer tissues and paired normal tissues. Using TRIzol reagent (Invitrogen, USA), we extracted the total RNA from the frozen tissues, which we then reverse-transcribed into cDNA with a PrimeScript RT reagent Kit with gDNA Eraser (TaKaRa). Next, we confirmed the expression of SLC2A1 with TB Green^®^ Premix Ex Taq (TaKaRa), following the manufacturer’s protocol, which we calculated using the 2−ΔΔCT method. We used GAPDH as the endogenous control. The primers used in this study were as follows: SLC2A1 forward 5′-CTGCAACGGCTTAGACTTCGAC-3′ and reverse 5′-TCTCTGGGTAACAGGGATCAAACA-3′; GAPDH forward 5′- GCTCTCTGCTCCTCCTGTTC-3′ and reverse 5′- ACGACCAAATCCGTTGACTC-3′.

### 2.2. Expression of SLC2A1 in Pan-Cancer

We extracted the expression data of SLC2A1 for each sample. We excluded cancer types with less than 3 samples. We used R software to calculate the expression differences between normal and tumor samples for each tumor by using Wilcoxon rank-sum and signed-rank tests. Moreover, we used the downloaded data to analyze the relationship between SLC2A1 level and clinicopathological parameters. We explored the protein level of SLC2A1 between human cancer and normal tissues by using the Human Protein Atlas (HPA: https://www.proteinatlas.org/) database. A previous study [16] from the Clinical Proteomic Tumor Analysis Consortium (CPTAC) identified 11 pan-cancer proteome-based subtypes (s1 to s11) using mass-spectrometry-based proteomic data from a compendium dataset of 2002 primary tumors compiled from 17 studies and 14 cancer types. The functions of proteome-based subtypes (s1 to s11) are described in detail in Appendix A. The UALCAN database (http://ualcan.path.uab.edu, accessed on 14 July 2022) provides a pan-cancer protein expression analysis option based on the data from CPTAC. Therefore, we used the UALCAN database to perform pan-cancer protein expression analysis of SLC2A1.

### 2.3. Diagnostic Value of SLC2A1 in Pan-Cancer

To explore whether the mRNA levels of SLC2A1 exhibit diagnostic efficiency for distinguishing cancer from normal lung tissues, we performed receiver operating characteristic (ROC) curve analysis for the TCGA-GTEx pan-cancer dataset. The pROC package was used to plot ROC curves and calculate the areas under the curves (AUCs) values in R.

### 2.4. Prognostic Analysis of SLC2A1

We used the Kaplan–Meier (log-rank) method and univariate Cox regression to evaluate the overall survival (OS) of the patients from the TCGA pan-cancer cohort. We also assessed the progression-free interval (PFI), disease-specific survival (DSS), and the disease-free interval (DFI) of the patients from the TCGA pan-cancer cohort with univariate Cox regression analysis. We determined the optimal cut-off value using the R package ‘survival’.

### 2.5. Genetic Alteration Analysis of SLC2A1

cBioPortal (http://cbioportal.org, accessed on 14 July 2022) is an open-access resource for exploring, visualizing, and analyzing multidimensional cancer genome data. It currently contained 225 cancer studies. We used cBioPortal to analyze the SLC2A1 gene genetic alterations in TCGA pan-cancer samples.

### 2.6. Epigenetic Analysis of SLC2A1

UALCAN database is a comprehensive, user-friendly, and interactive web resource for analyzing cancer OMICS data. We used UALCAN to evaluate promoter methylation of SLC2A1 in pan-cancer.

We collected 21 genes related to RNA m6A methylation from previous studies [17]. We extracted the SLC2A1 gene expression and 21 RNA m6A-methylation-related genes’ expression data from each sample in the TCGA pan-cancer dataset. Then, we analyzed the correlation between SLC2A1 and RNA m6A-methylation-related genes in pan-cancer, and the results are presented in a heatmap.

### 2.7. Functional Enrichment Analysis of SLC2A1

We selected the TCGA LUAD cohort as an example to explore the underlying mechanisms of SLC2A1. Based on the median expression of SLC2A1, we divided the patients into high and low groups. After that, we conducted Gene Ontology (GO), Kyoto Encyclopedia of Genes and Genomes (KEGG), and Gene Set Enrichment Analysis (GSEA) (www.gsea-msigdb.org/gsea/index.jsp, accessed on 14 July 2022). First, we used the “limma” package in R to screen differential expression genes (DEGs) between these two groups. We set FDR<0.05 and |log2FC|≥ 1 as the threshold values for DEG identification. After that, the enrichGO and enrichKEGG functions of the ClusterProfiler package in Bioconductor were used to perform GO/KEGG analysis on SLC2A1-related DEGs, choosing p.adj < 0.05, *q*-value < 0.05, and count ≥2 as cut-off values. Second, we performed GSEA based on the HALLMARK and REACTOME gene sets. Under the condition of FDR (*q*-value) < 0.25 and *p* < 0.05, the results were considered statistically significant. In addition, we used the single-cell database CancerSEA (http://biocc.hrbmu.edu.cn/CancerSEA/, accessed on 14 July 2022) to study the potential functions of SLC2A1. The aim of the CancerSEA database is to help researchers better understand various functional states of cancer cells at the single-cell level. This database contained 41,900 cancer single cells from 25 cancers, a total of 280 cell groups, and summarized 14 functional statuses of cancer cells.

### 2.8. Pan-Cancer Analysis of Correlation of SLC2A1 Expression with Tumor Cell Infiltration

TME plays an important role in the occurrence and development of cancers. First, we used three algorithms, ESTIMATEScore, MCP-counter score, and EPIC [18,19,20], to evaluate the tumor immune infiltration in pan-cancer from the TCGA dataset via the SangerBox website (http://vip.sangerbox.com/home.html, accessed on 14 July 2022), which is a useful online platform for TCGA data analysis. Second, we compared the differences in ImmuneScore, StromalScore, and ESTIMATEScore between patients from the low-SLC2A1 and high- SLC2A1 groups with Wilcoxon signed-rank test. In addition, Spearman’s correlation analysis was used to evaluate the relationship between SLC2A1 and the tumor immune infiltration evaluated by the MCP counter score and EPIC algorithms.

### 2.9. Correlation between SLC2A1 and Immune Checkpoint Genes, Tumor Mutation Burden (TMB), and Microsatellite Instability (MSI) in Pan-Cancer

According to a previous study [21], we collected 60 immune checkpoint (ICP) genes, which included 36 immune stimulators and 24 immune inhibitors among. Using the SangerBox tools, we analyzed the correlation between SLC2A1 expression and ICP genes. TMB [22] and MSI [23] are effective biomarkers for cancer immunotherapy. The correlations between SLC2A1 expression and TMB and MSI were also explored via the SangerBox website.

### 2.10. Statistical Analysis

We used R version 4.1.0 to perform the statistical analysis. Survival analysis was carried out according to Kaplan–Meier analysis, the log-rank test, and Cox regression analysis. We compared the continuous variables using Student’s t-test or the Wilcoxon rank-sum test, as appropriate. Categorical clinicopathological variables were compared using the chi-square test or Fisher’s exact test. Correlation analysis was performed by Pearson correlation analysis. A *p*-value of less than 0.05 was considered statistically significant (ns, *p* ≥ 0.05; *, *p* < 0.05; **, *p* < 0.01; ***, *p* < 0.001; ****, *p* < 0.0001).

## 3. Results

### 3.1. Pan-Cancer Expression Landscape of SLC2A1

To preliminarily understand the expression of SLC2A1 in cancers, we first evaluated SLC2A1 mRNA expression in the TCGA-GTEx pan-cancer dataset. The results revealed that SLC2A1 expression was significantly upregulated in 22 cancer types: ACC, BLCA, BRCA, CESC, CHOL, COAD, ESCA, GBM, HNSC, KIRC, LGG, LIHC, LUAD, LUSC, OV, PAAD, READ, STAD, TGCT, THCA, UCEC, and UCS. In comparison, low SLC2A1 expression was observed in five kinds of tumors: DLBC, KICH, LAML, SKCM, and THYM (Figure 1). For paired tumor and normal tissues in TCGA pan-cancer, SLC2A1 levels was were significantly higher in BRCA, CHOL, COAD, ESCA, KIRC, LIHC, LUAD, LUSC, READ, STAD, THCA, and UCEC, but lower in KICH and PRAD (Appendix A).

To further validate the differential mRNA expression of SLC2A1, we comprehensively searched the GEO database and found a total of 20 relative datasets for the validation of the SLC2A1 pan-cancer analysis. As shown in Figure 2A–T, we confirmed that SLC2A1 was significantly highly expressed in 19 cancer types: ACC, BLCA, BRCA, CESC, CHOL, COAD, ESCA, HNSC, KIRC, LIHC, LUAD, LUSC, OV, PAAD, READ, STAD, TGCT, THCA, and UCEC; SLC2A1 expression was significantly lower in SKCM. Furthermore, we collected 30 pairs of samples with LUAD, 30 pairs of samples with COAD, and 30 pairs of samples with ESCA in the Sixth Affiliated Hospital of Sun Yat-Sen University. We detected the expression of SLC2A1 in the paired samples by qPCR. The results showed that the expression of SLC2A1 in LUAD (*p* < 0.0001), COAD (*p* < 0.0001), and ESCA (*p* < 0.0001) tissues was much higher than that in paired normal tissues (Figure 2U–W). The above results strongly suggested that SLC2A1 is overexpressed in most cancer tissues.

### 3.2. Association between SLC2A1 Expression and Clinicopathologic Parameters in Pan-Cancer

To explore the association between SLC2A1 expression and the clinicopathologic parameter of cancers, we performed differential analysis of SLC2A1 expression among different pathological stages of patients in pan-cancer. The results revealed that the expression of SLC2A1 was significantly higher in higher stages in most tumors, including ACC, CESC, COAD, COADREAD, PAAD, KIRP, LIHC, LUAD, TGCT, UCEC, and UVM (Figure 3A–K). The above results indicated that the expression of SLC2A1 is higher as pathological stage advances in most cancers

### 3.3. Protein Level Analysis of SLC2A1

The previous results confirmed that *SLC2A1* is highly expressed in most cancers at the mRNA level, but whether *SLC2A1* is also highly expressed at the protein level needed further exploration. We used the HPA database to verify the protein expression level of SLC2A1. The subcellular localization of SLC2A1 in cancer cells indicated that it is predominantly expressed in the plasma membrane (Figure 4A). The HPA database included 20 types of immunohistochemistry data on cancers. We found that SLC2A1 was strongly or medium stained in most cancers, but was negative or weakly stained in most normal tissues. The detailed information was as follows: lung (weak) vs. LUAD (strong), liver (negative) vs. LIHC (strong), testis (weak) vs. TGCT (strong), cervix (negative) vs. CESC (strong), thyroid (weak) vs. THCA (medium), colon (negative) vs. COAD (strong), ovary (weak) vs. OV (strong), brain (negative) vs. GBMLGG (strong), bladder (negative) vs. BLCA (strong), skin (weak) vs. SKCM (strong), pancreas (weak) vs. PAAD (strong), breast (negative) vs. BRCA (strong), kidney (weak) vs. KIRC (strong), tongue (negative) vs. HNSC (strong), and stomach (weak) vs. STAD (strong) (Figure 4B–P). The above results indicated that SLC2A1 is highly expressed in most cancers at the protein level. In addition, we analyzed the protein expression of SLC2A1 of 2002 patients across 14 cancer subtypes in the CPTAC samples based on UALCAN data. In the CPTAC samples, we found 11 proteome-based subtypes (s1–s11), and the statistical results between 2 of the 11 subtypes are shown in Appendix A. High SLC2A1 expression strongly correlated with proteome-based subtype s8 (Figure 5). These findings suggested that SLC2A1 may have an important regulatory role in the progression of various cancers and may be related to the immune system process, extracellular region, and glycolysis.

### 3.4. Diagnostic Value of SLC2A1 in Pan-Cancer

Although we found that SLC2A1 is highly expressed in cancers compared with in normal tissues, whether SLC2A1 has diagnostic value for cancers still need further analysis. We evaluated the diagnostic value of SLC2A1 in pan-cancer by using ROC curves. AUC > 0.7 is considered high accuracy [24]. The results identified 24 cancer types (AUC > 0.7): ACC (AUC = 0.751), BRCA (AUC = 0.820), CESC (AUC = 0.814), COAD (AUC = 0.968), COADREAD (AUC = 0.962), ESCA (AUC = 0.841), GBM (AUC = 0.821), GBMLGG (AUC = 0.775), KIRC (AUC = 0.893), LAML (AUC = 0.929), LGG (AUC = 0.762), LUAD (AUC = 0.917), LUSC (AUC = 0.996), HNSC (AUC = 0.903), OV(AUC = 0.973), PAAD (AUC = 0.986), READ (AUC = 0.975), SKCM (AUC = 0.850), STAD (AUC = 0.903), TGCT (AUC = 0.960), THCA (AUC = 0.745, THYM (AUC = 0.732), UCEC (AUC = 0.865), and UCS (AUC = 0.889) (Figure 6A–X). Notably, SLC2A1 had very high accuracy in predicting COAD, COADREAD, LAML, LUAD, LUSC, HNSC, OV, PAAD, READ, STAD, and TGCT (AUC > 0.9). These results suggested that SLC2A1 may have valid pan-cancer diagnostic value.

### 3.5. Prognostic Value of SLC2A1 in Pan-Cancer

Whether the high expression of SLC2A1 in cancers affects the prognosis of patients is an issue of concern to researchers. We used two methods, Kaplan–Meier and univariate Cox regression analyses, to evaluate the prognostic value of SLC2A1 in pan-cancer. First, the results of Kaplan–Meier analysis showed that SLC2A1 was a hazard factor for the OS of patients with ACC, BLCA, CESC, GBMLGG, HNSC, KICH, KIPAN, KIRP, LGG, LIHC, LUAD, MESO, OV, PAAD, SARC, SKCM, SKCM-M, and THYM (Figure 7A–R). Second, we used univariate Cox regression analysis to evaluate the OS, PFI, DSS, and DFI of the patients. The results of OS analysis revealed that SLC2A1 acted as a hazard factor for patients with LIHC, LUAD, KIRP, MESO, ACC, PAAD, KICH, SARC, CESC, BLCA, and SKCM (Figure 8A). The results of PFI analysis showed that SLC2A1 acted as a hazard factor for patients with ACC, KICH, KIRP, PAAD, LUAD, MESO, SARC, and BLCA (Figure 8B). The results of DSS analysis indicated that SLC2A1 acted as a hazard factor for patients with KIRP, LUAD, PAAD, MESO, ACC, KICH, LIHC, SARC, and BLCA (Appendix A). The results of DFI analysis showed that SLC2A1 acted as a hazard factor for patients with PAAD, LUAD, COAD, ACC, MESO, KIRC, and TGCT (Appendix A). The above results suggested that patients with high expression of SLC2A1 have a poor prognosis in most cancers.

### 3.6. Genetic Alteration Analysis of SLC2A1

The above results indicated that SLC2A1 is highly expressed in most cancers, and carries a poor prognosis. Genetic alteration is one of the key factors that affects gene expression [25]. Thus, we analyzed he genetic alteration status of SLC2A1 in the TCGA pan-cancer cohorts. We included a total of 10,967 pan-cancer patients in the cBioPortal in the study. OncoPrint showed that the overall genetic alteration rate of SLC2A1 in cancers was relatively low (only 1.8%) (Appendix A). As shown in Appendix A, 24 of 32 types of cancer had *SLC2A1* gene alteration data. The highest alteration frequency of SLC2A1 appeared in the ovarian carcinoma patients with “amplification” as the primary type. Additionally, amplification was the main genetic alteration type in some other cancers, such as BLCA, ESCA, SARC, LUSC, BRCA, LUAD, MESO, ACC, and LIHC, whose frequency ranged from 1% to 4%. The types and sites of the SLC2A1 mutations are further presented in Appendix A. The results showed 80 mutation sites in the SLC2A1 gene, and the missense mutation of SLC2A1 was the main type of genetic mutation. These findings suggested that the genetic alteration status of SLC2A1 may not be the cause of the high expression of SLC2A1 in cancer tissues.

### 3.7. Epigenetic Analysis of SLC2A1

Epigenetic modifications, such as DNA promoter methylation and RNA m6A methylation, regulate the gene expression, thus affecting the growth and development of cancers [26]. Therefore, to explore the cause of the high expression level of SLC2A1 in cancers, we analyzed DNA promoter methylation and RNA methylation. First, we investigated DNA promoter methylation of SLC2A1 in pan-cancer by using the UALCAN database. We used 24 types of cancers in the UALCAN database to analyze the methylation of SLC2A1. The results showed that DNA methylation significantly differed in nine types of cancers compared with normal tissues. We observed a significant decrease in the methylation level of SLC2A1 in BLCA, KIRC, LIHC, LUAD, LUSC, UCEC, TGCT, and THCA (Figure 9A,C–I), and a significant increase in the level in COAD (Figure 9B). The above results suggested that SLC2A1 gene promoter methylation may be one of the reasons for the high expression level of SLC2A1 in some cancers. Moreover, we explored the correlation between SLC2A1 and m6A-methylation-related genes in pan-cancer, and the results demonstrated that most of the m6A-methylation-related genes positively correlated with the expression of SLC2A1 in 33 TCGA cancers, which suggested that m6A methylation plays an important role in the epigenetic modification of SLC2A1 (Figure 10).

### 3.8. Functional Enrichment Analysis of SLC2A1

To comprehensively explore the mechanism underlying SLC2A1 leading to the poor prognosis of cancer patients, we used LUAD as an example to perform GO, KEGG, and GSEA analyses. We regarded the median value of SLC2A1 as the cut-off point. First, during GO and KEGG analyses, we detected 346 DEGs (FDR < 0.05 and |log2FC| ≥ 1), of which 154 genes were downregulated and 192 genes were upregulated (Appendix A). The heat map showed the top 50 upregulated and downregulated DEGs related to SLC2A1 (Appendix A). Under the condition of p.adj < 0.05, *q*-value < 0.05, and count ≥ 2, we found that SLC2A1-related DEGs are involved in 269 biological process (GO-BP), 56 in cell component (GO-CC), 10 in molecular function (GO-MF), and 14 in KEGG (Appendix A). The bubble graph demonstrated the top 10 messages for GO-BP, GO-CC, GO-MF, and KEGG (Figure 11A–D). The GO functional annotations showed that SLC2A1-related DEGs are mainly involved in the cell-cycle regulation, neutrophil mediated immunity, neutrophil activation, etc. The results of KEGG pathway analysis demonstrated that SLC2A1-related DEGs are primarily associated with cell cycle, glycolysis/gluconeogenesis, carbon metabolism, etc. Second, using the REACTOME and HALLMARK gene sets, we performed GSEA to identify the functional enrichment of high and low SLC2A1 expression. The results of GSEA with |NES| >1, p.adj < 0.05 and q-value (FDR) < 0.25 are shown in Appendix A. The HALLMARK enrichment term showed that SLC2A1 is mainly associated with epithelial–mesenchymal transition (NES = 2.235, P = 0.003, FDR < 0.001), glycolysis (NES=2.167, P=0.003, FDR<0.001), and hypoxia (NES = 2.059, P = 0.003, FDR < 0.001), (Figure 12A–C). The REACTOME enrichment term showed that SLC2A1 is mainly associated with cell-cycle checkpoints (NES = 2.455, P = 0.010, FDR = 0.006), mitotic metaphase and anaphase (NES = 2.410, P = 0.011, FDR = 0.006), and DNA repair (NES = 2.244, P = 0.011, FDR = 0.006) (Figure 12D–F). In addition, single-cell analysis can provide a profound understanding of the biological characteristics of cancer. We analyzed the correlation between SLC2A1 and 14 functional states in pan-caner by using the single-cell CancerSEA database. The results showed that SLC2A1 is mainly positively related to hypoxia, angiogenesis, epithelial–mesenchymal transition (EMT), and metastasis, and negatively related to DNA repair (Figure 12G). Collectively, the above results suggested that SLC2A1 affects the growth and development of cancers through multiple mechanisms, including immune regulation.

### 3.9. Immune Cell Infiltration Analysis of SLC2A1

The above results of function enrichment analysis and the protein expression of SLC2A1 across pan-cancer subtypes in the CPTAC samples suggested that SLC2A1 may be related to immune regulation, so we conducted an immune cell infiltration analysis, We used three algorithms (ESTIMATEscore, EPIC, and MCPcounter) to explore the relationship between SLC2A1 expression and immune cell infiltration of TME. First, using the ESTIMATEscore, we found high expression levels of SLC2A1 are related to low ImmuneScore, low StromalScore, and low ESTIMATEScore in most cancers, such as ACC, BLCA, BRCA, ESCA, HNSC, LUAD, OSCC, PAAD, PRAD, SARC, SKCM, STAD, TGCT, UCEC, and UCS (Appendix A). Second, using the EPIC algorithm, we found that SLC2A1 expression is positively correlated with cancer-associated fibroblasts (CAFs) in most cancers. SLC2A1 expression is negatively correlated with CD8+ T cells in 12 types of cancers (STES, TGCT, ESCA, LUSC, SKCM, LUAD, BLCA, HNSC, CESC, LAMLC, THYM, and GBM), but positively with CD8+ T cells in 5 types of cancers (PRAD, KIPAN, KIRP, CHOL, and LIHC) (Figure 13A). Third, using the MCPcounter algorithm, we found that SLC2A1 expression is positively correlated with neutrophils and CAFs in most cancers. SLC2A1 expression is negatively correlated with CD8+ T cells in 12 types of cancers (LUSC, TGCT, THYM, HNSC, BRCA, SKCM, STES, GBMLGG, GBM, PAAD, ALL, and ESCA), but positively correlated with CD8+ T cells in 5 types of cancers (KIPAN, LIHC, LAML, PCPG, and CHOL) (Figure 13B). The above results indicated that SLC2A1 has an impact on the infiltration of immune cells in the TME of most cancers and is especially positively correlated with neutrophils and CAFs in the TME.

### 3.10. SLC2A1 Related to Immune Checkpoint (ICP) Genes, TMB, and MSI in Human Cancers

Immune surveillance affects the growth and development of cancer cells, and cancer cells evade immune responses by taking advantage of ICP [27]. ICP genes are divided into two major categories: immunoinhibitors and immunostimulators. As such, we investigated the associations between SLC2A1 expression and the two main types of immune modulators in human cancers to explore the potential function of SLC2A1 in immunotherapy. The results showed a certain correlation between SLC2A1 and immune modulators in all 33 tumor types. We found that the expression of SLC2A1 is positively correlated with most immunoinhibitors and immunostimulators in LAML, LIHC, UVM, THCA, PCPG, PRAD, OV, READ, KIRC, KIPAN, DLBC, and THYM. In contrast, the expression of SLC2A1 is negatively correlated with most immunoinhibitors and immunostimulators in TGCT, ESCA, STES, HNSC, and LUSC. Notably, SLC2A1 is remarkably positively correlated with CD274 (PD-L1) and CTLA4 in most cancers (Figure 14). These findings suggested that SLC2A1 may affect the immune checkpoint blockade treatment response in human cancers.

TMB and MSI are two new biomarkers that reflect the response of immunotherapy. So, we explored the correlation between SLC2A1 expression and TMB, and MSI. The expression of SLC2A11 is significantly positively correlated with TMB in most cancers, including PAAD, ACC, LUAD, THYM, GBM, SARC, STAD, CESC, BRCA, GBMLGG, STES, and HNSC, but negatively correlated with TMB in SKCM (Figure 15A). We also investigated the correlation of the SLC2A1 expression with MSI in pan-cancer: ACC, UVM, TGCT, SARC, STAD, and STES exhibited positive correlations; DLBC, KIPAN, GBMLGG, and PRAD exhibited negative correlations (Figure 15B). The above results indicated that SLC2A1 may be used to predict the response to immunotherapy.

## 4. Discussion

The increase in glycolysis is one of the main characteristics of glycometabolism in cancer cells [28]. The *SLC2A1* gene is one of the key genes in cancer glycometabolism, which promotes the glycolysis of cancer cells, thus affecting their growth and metastasis [29,30]. Studies on SLC2A1 in pan-cancer analysis were previously lacking. This is the first study in which the role of SLC2A1 at the pan-cancer level was explored.

In our study, based on the TCGA-GTEx pan-cancer dataset, we found that SLC2A1 is highly expressed in most cancers compared with normal tissues, which we further confirmed via the GEO datasets, protein expression data from the HPA database, and our own data. Additionally, the expression of SLC2A1 is significantly upregulated in higher pathologic stages in various cancers. Previous studies have reported that SLC2A1 is highly expressed in a variety of cancers, such as BLCA, LUAD, LIHC, CESC, COAD, OV, UCEC, BRCA, STAD, ESCA, and PAAD [11,12,13,14,15,31,32,33,34,35,36,37]. These results are consistent with our research results. In addition, the results of the ROC analysis of pan-cancers revealed that the AUC values of most cancers were greater than 0.7. Therefore, our results suggested that SLC2A1 may play an important role in the growth and metabolism of cancers, and they support the possibility of SLC2A1 being used as a biomarker for the diagnosis of pan-cancer.

The evaluation of the prognostic value of SLC2A1 is an indispensable part of our study. We assessed the prognostic value of SLC2A1 in pan-cancer by using Kaplan–Meier and Cox regression analyses. The results showed that SLC2A1 is a hazard factor for the OS of patients in most cancers. Mortality from noncancerous causes may not reflect tumor biology, aggressiveness, or response to therapy. Additionally, a longer follow-up time is required for using OS. Therefore, to more accurately reflect the impact of SLC2A1 on the prognosis of patients, we further conducted a univariate COX analysis for PFI, DFI, and DSS of patients. The results also showed that SLC2A1 is a hazard factor for the PFI, DFI and DSS of patients in various cancers. These results indicated that high SLC2A1 expression mainly plays a hazardous role in patient prognosis for most cancer types. The high expression of SLC2A1 in cancers is often associated with poor prognosis [38,39].

To study why SLC2A1 is highly expressed in pan-cancer, we performed genetic alteration, DNA promoter methylation, and RNA m6A methylation analyses. We found that the overall mutation rate of SLC2A1 in pan-cancer is only 1.8%, which could not explain the high transcriptome expression of SLC2A1 in most cancers. Klepper et al. reported that Glut1 deficiency syndromes are due to SLC2A1 genetic variation [40]. Therefore, we think that genetic alteration is not the main cause of the high expression of SLC2A1. In addition, DNA promoter methylation can regulate gene expression without changing the DNA sequence, which is one of the main forms of DNA epigenetic modification [41]. A higher level of DNA promoter methylation means lower expression of the corresponding gene [42]. We found hypomethylation in the SLC2A1 promoter region in various cancer tissues, which might, to some extent, explain the SLC2A1 mRNA overexpression in the corresponding cancers. Furthermore, RNA m6A methylation is an important mechanism affecting the regulation of RNA expression. We collected 21 RNA m6A-methylation-related genes and performed pan-cancer correlation analysis between SLC2A1 and m6A-methylation-related genes, finding that they are significantly correlated in pan-cancer. This indicated that the mechanism of m6A methylation may play a vital role in the regulation of SLC2A1 expression in cancer tissues. Shen et al. demonstrated that the m6A-IGF2BP2/3-dependent mechanism inhibits the mRNA degradation of SLC2A1 in colorectal cancer [43]. Some other mechanisms may lead to the overexpression of SLC2A1, for example, histone modification on chromosomes [44] or RNA m1A/m5C modification [45]. We did not consider these potential mechanisms in this study.

The mechanism of SLC2A1 in pan-cancer has rarely been explored. LUAD is one of the most common cancer types. In our study, we found that SLC2A1 is highly expressed in LUAD, and was related to a poor prognosis in terms of OS, PFI, DSS, and DFI, so we took LUAD as the representative tumor in the GO, KEGG, and GSEA analyses of SLC2A1. The results showed that SLC2A1 is mainly associated with hypoxia, EMT, glycolysis, cell-cycle regulation, DNA repair, and neutrophil-mediated immunity. Single-cell sequencing can help us better understand the biological features of cancers, to achieve the goal of more precise cancer treatment [46,47]. In our study, we used the single-cell database CancerSEA to explore the function of SLC2A1 in pan-cancer. We discovered that SLC2A1 is remarkably positively related to hypoxia, EMT, and metastasis, and negatively related to DNA repair, which was roughly consistent with the bulk analysis. Most of these functions of SLC2A1 have been studied in various types of cancers. For example, hypoxia induces GLUT1 overexpression in various cancers [48]; Azadeh Nilchian et al. revealed that overexpression of GLUT1 can make breast cancer cells produce stable EMT and promote proliferation during chronic TGF-β1 exposure [49]; Zhang et al. revealed that GLUT1 S-palmitoylation mediated by DHHC9 promotes glycolysis and tumorigenesis in glioblastoma [50]. Takahashi et al. reported that Glut-1 knockdown also induces cell-cycle arrest in pancreas cancer cells [51]; Kim et al. found that increased GLUT1 expression can repair damaged DDR in SALL4-deficient human cancer cells [52]. In summary, SLC2A1 may be overexpressed in cancers tissues by hypoxia and promote cancer-cell proliferation and metastasis by regulating cancer glycometabolism, the cell-cycle checkpoint, DNA repair process, and EMT.

The TME plays a key role in the support of tumor progression, invasiveness, and metastasis [53]. In our study, we found that high SLC2A1 expression strongly correlates with proteome-based subtypes s8. A previous study [16] demonstrated that s8 is related to the immune system process, extracellular region, and glycolysis. Therefore, we explored the relationship of SLC2A1 expression with immune cell infiltration by using three algorithms (ESTIMATEScore, EPIC, and MCPcounter). We found that high expression of SLC2A1 corresponds to low immune, stromal and ESTIMATE scores in the TME in most cancer types, which indicated that SLC2A1 might be mainly expressed by cancer cells. In addition, SLC2A1 is positively correlated with the neutrophils and CAFs in the TME in most cancers. Neutrophils and CAFs in the TME promote tumor proliferation and metastasis [54,55,56]. Thus, SLC2A1 may promote tumor proliferation and metastasis by affecting neutrophils and CAFs in the TME. Ancey et al. revealed that the radiotherapy resistance of lung cancer depends on GLUT1-mediated glucose uptake in tumor-associated neutrophils [54]. Sun et al. discovered that phosphorylating GLUT1 enhances glycolytic activity of the CAFs in breast cancer to promote cancer-cell invasion by activating the TGFβ1/p38 MAPK/MMP2/9 signaling axis [57]. CD8+ T cells are an important component of cancer immunity. Our findings showed that SLC2A1 is positively correlated with CD8+ T cells in some cancer types, but negatively correlated in other cancer types, suggesting that the effect of SLC2A1 on CD8+ T cells in the TME is complex.

ICP genes affect the immune cell infiltration of the TME and cancer immunotherapy [58]. As such, we analyzed the relationship between the SLC2A1 expression and ICP genes. The results showed a certain correlation between SLC2A1 and ICP genes in all 33 tumors. This indicated that SLC2A1 has an impact on immune function in the TME. Notably, SLC2A1 is remarkably positively correlated with CD274 (PD-L1) and CTLA4 in most cancers. Young Wha Koh et al. demonstrated that PD-L1 protein and mRNA expressions are correlated with GLUT1 expression in lung adenocarcinoma [59]. PD-L1 and CTLA4 are the essential biomarkers of T cell exhaustion [60]. The definition of T-cell exhaustion is that the T-cell functions of patients with common chronic infection or cancer is impaired or even lost. Therefore, the expression of SLC2A1 results in the T-cell exhaustion of the TME. MSI and TMB are the biomarkers for predicting cancer immunotherapy response [22,23]. We performed a correlation analysis between SLC2A1 and TMB and MSI, and we found that SLC2A1 expression is significantly correlated with TMB in 13 cancer types and with MSI in 10 cancer types. Additionally, PD-L1 and CTLA4 are the main targets of clinical immunotherapy. This indicated that SLC2A1 may be considered as a biomarker to predict the response to immunotherapy.

The limitations in our study should be considered when generalizing the findings. First, our results were mainly generated from bioinformatics analysis. In vivo and in vitro experiments are needed to prove our results regarding the potential function of SLC2A1. Second, the microarray and sequencing data from different databases might have caused systematic bias. Third, the data used in this study were retrospective. Our results should thus be further confirmed by prospective studies in the future. Finally, no anti-SLC2A1 therapeutic monoclonal antibodies have yet been evaluated in clinical trials. Therefore, we have no specific and complete cases with data to identify the benefit of anti-SLC2A1-targeting drugs in the survival of cancer.

## 5. Conclusions

SLC2A1 may play a crucial role in cancer cell proliferation and metastasis through multiple mechanisms, such as regulating the cell-cycle checkpoint, DNA repair process, EMT, CAFs and neutrophils in the TME, and T-cell exhaustion. SLC2A1 might serve as a novel pan-cancer diagnostic and prognostic biomarker and provide an opportunity to develop new immunotherapy strategies.

## Figures and Tables

**Figure 1 cancers-14-05344-f001:**
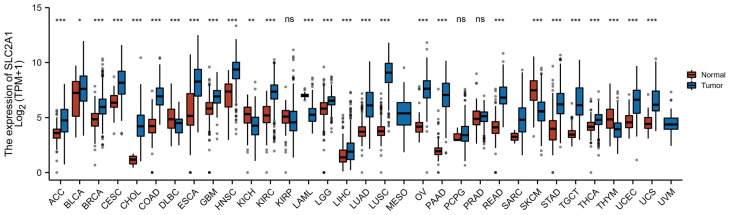
SLC2A1 expression in pan-cancer. ACC, adrenocortical carcinoma; BLCA, bladder urothelial carcinoma; BRCA, breast invasive carcinoma; CESC, cervical squamous cell carcinoma and endocervical adenocarcinoma; CHOL, cholangiocarcinoma; COAD, colon adenocarcinoma; DLBC, lymphoid neoplasm diffuse large B-cell lymphoma; ESCA, esophageal carcinoma; GBM, glioblastoma multiforme; HNSC, head and neck squamous cell carcinoma; KICH, kidney chromophobe; KIRC, kidney renal clear cell carcinoma; KIRP, kidney renal papillary cell carcinoma; LAML, acute myeloid leukemia; LGG, brain lower grade glioma; LIHC, liver hepatocellular carcinoma; LUAD, lung adenocarcinoma; LUSC, lung squamous cell carcinoma; MESO, mesothelioma; OV, ovarian serous cystadenocarcinoma; PAAD, pancreatic adenocarcinoma; PCPG, pheochromocytoma and paraganglioma; PRAD, prostate adenocarcinoma; READ rectum adenocarcinoma; SARC, sarcoma; SKCM, skin cutaneous melanoma; STAD, stomach adenocarcinoma; TGCT, testicular germ cell tumor; THCA, thyroid carcinoma; THYM, Thymoma; UCEC, uterine corpus endometrial carcinoma; UCS, uterine carcinosarcoma; UVM uveal melanoma(ns, *p* ≥ 0.05; *, *p* < 0.05; **, *p* < 0.01; ***, *p* < 0.001).

**Figure 2 cancers-14-05344-f002:**
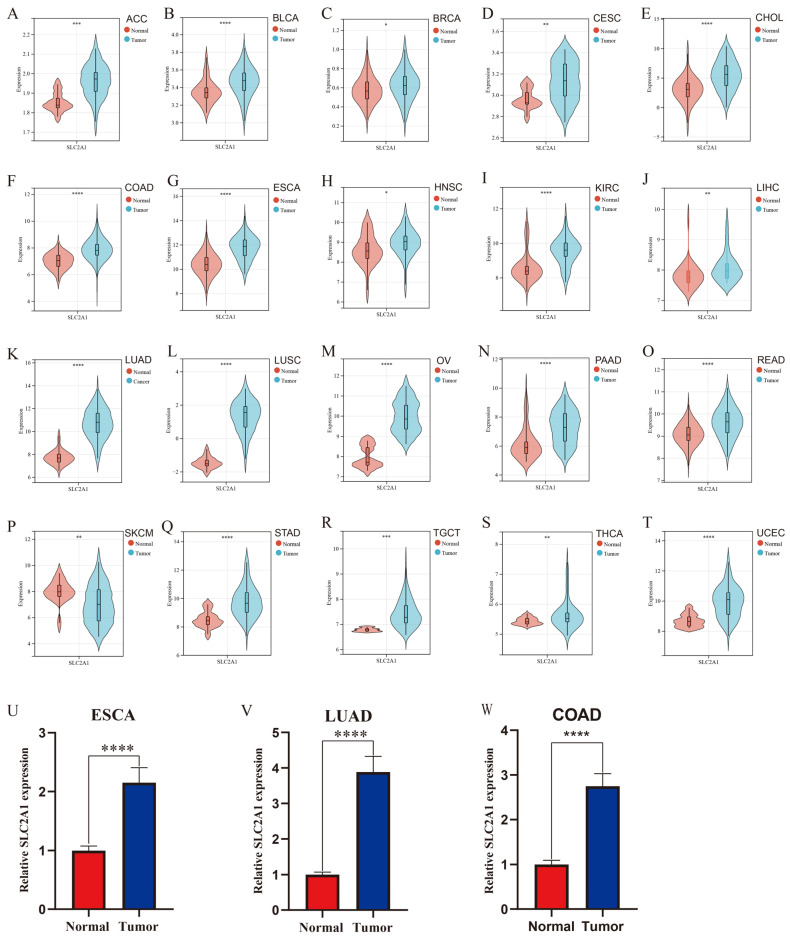
Validation of expression of SLC2A1 in pan-cancer. We used 20 GEO datasets for validation (**A**–**T**); SLC2A1 expression in 30 pairs of ESCA tissues and their normal counterparts was measured by qPCR (**U**); SLC2A1 expression in 30 pairs of LUAD tissues and their normal counterparts was measured by qPCR (**V**); SLC2A1 expression in 30 pairs of COAD tissues and their normal counterparts was measured by qPCR (**W**) (*, *p* < 0.05; **, *p* < 0.01; ***, *p* < 0.001; ****, *p* < 0.0001).

**Figure 3 cancers-14-05344-f003:**
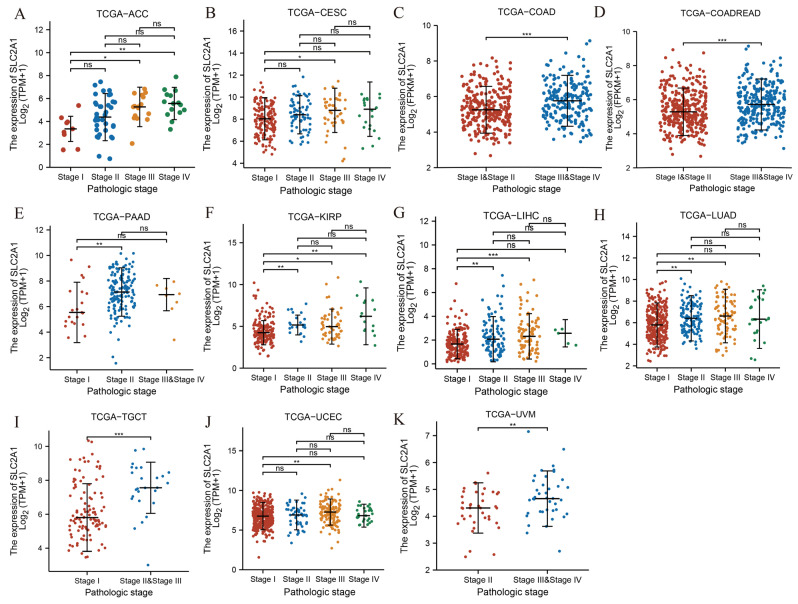
Pan-cancer differential expression of SLC2A1 in different pathologic stages in indicated tumor types from TCGA database (**A**–**K**) (ns, *p* ≥ 0.05; *, *p* < 0.05; **, *p* < 0.01; ***, *p* < 0.001).

**Figure 4 cancers-14-05344-f004:**
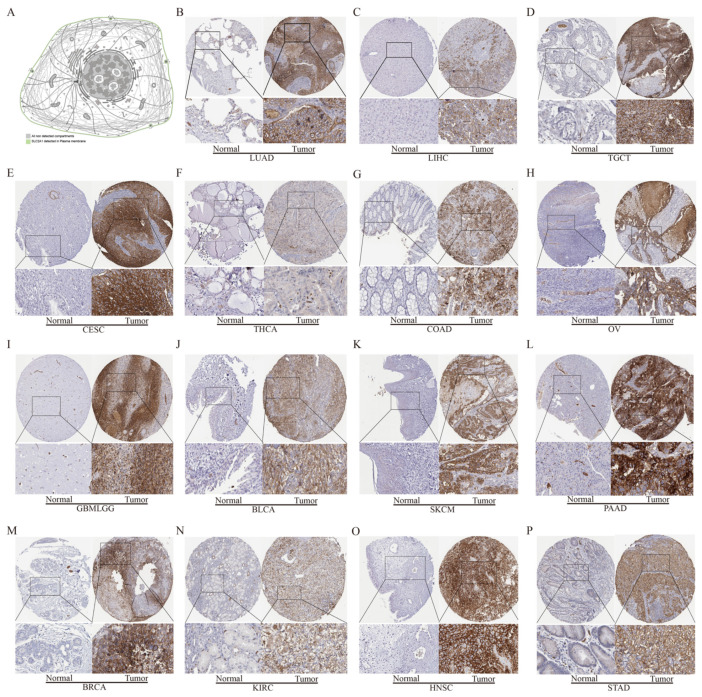
Protein level analysis of SLC2A1 in pan-cancer. Subcellular localization of SLC2A1 in cancer cells per the HPA database (**A**); immunohistochemical data of SLC2A1 in pan-cancer from HPA dataset (**B**–**P**).

**Figure 5 cancers-14-05344-f005:**
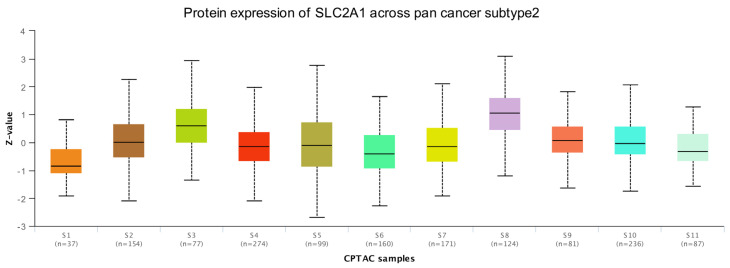
Protein expression of SLC2A1 across pan-cancer subtype in CPTAC samples based on UALCAN data.

**Figure 6 cancers-14-05344-f006:**
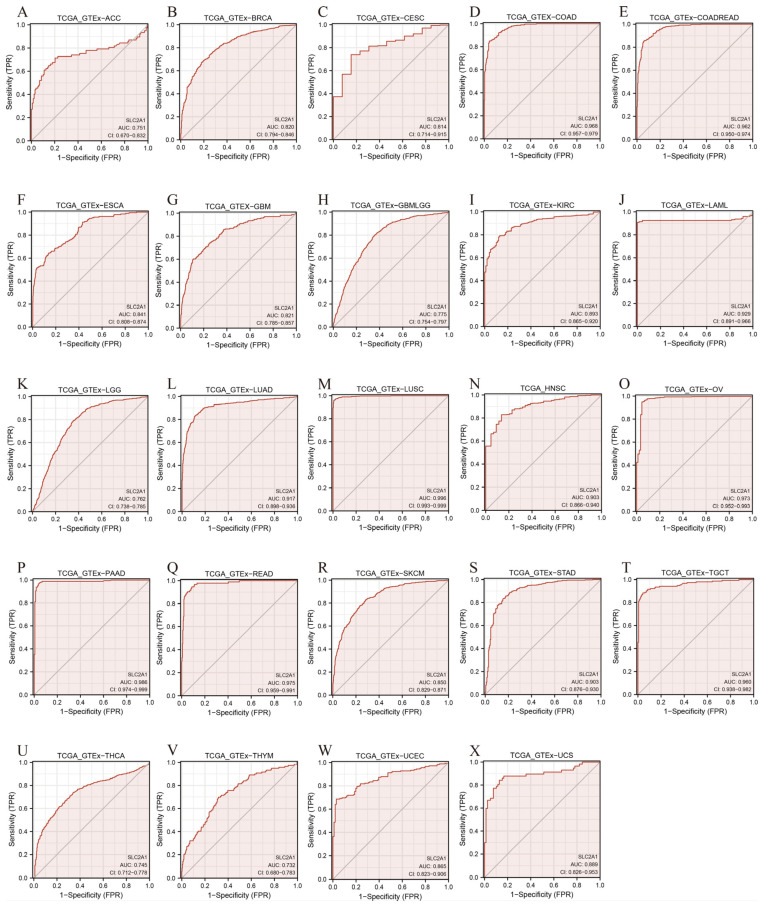
Receiver operating characteristic (ROC) curve for SLC2A1 expression in pan-cancer (**A**–**X**).

**Figure 7 cancers-14-05344-f007:**
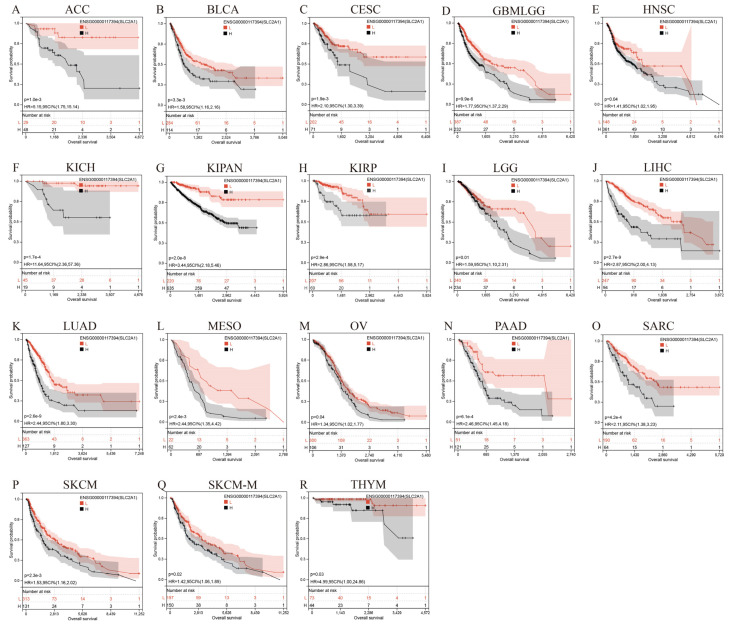
Pan-cancer Kaplan–Meier overall survival of SLC2A1 in indicated tumor types from TCGA database (**A**–**R**).

**Figure 8 cancers-14-05344-f008:**
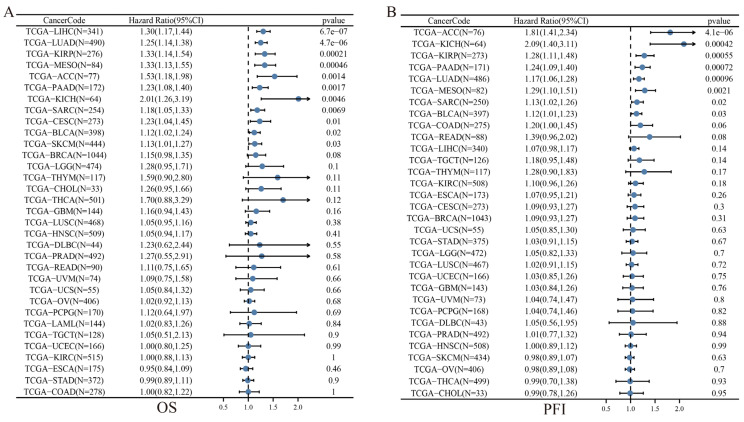
Univariate Cox regression analysis of SLC2A1. Forest map shows univariate Cox regression results of SLC2A1 for OS (**A**) and PFI (**B**) in pan-cancer.

**Figure 9 cancers-14-05344-f009:**
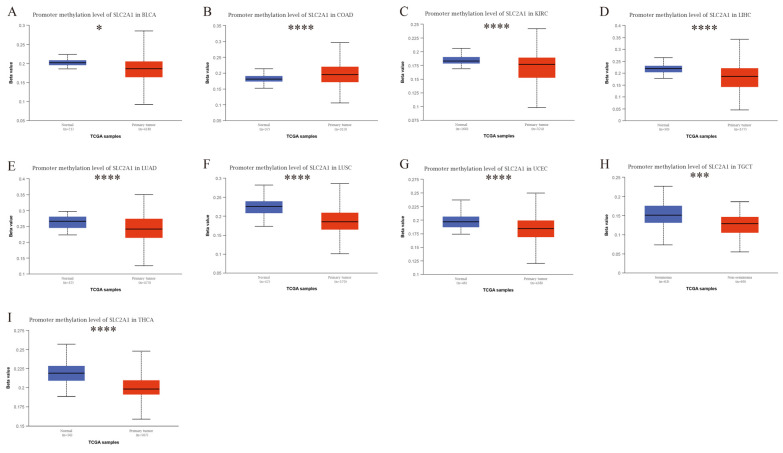
Promoter methylation level of SLC2A1 in pan-cancer: BLCA (**A**), COAD (**B**), KIRC (**C**), LIHC (**D**), LUAD (**E**), LUSC (**F**), UCEC (**G**), TGCT (**H**), and THCA (**I**) (*, *p* < 0.05; ***, *p* < 0.001; ****, *p* < 0.0001).

**Figure 10 cancers-14-05344-f010:**
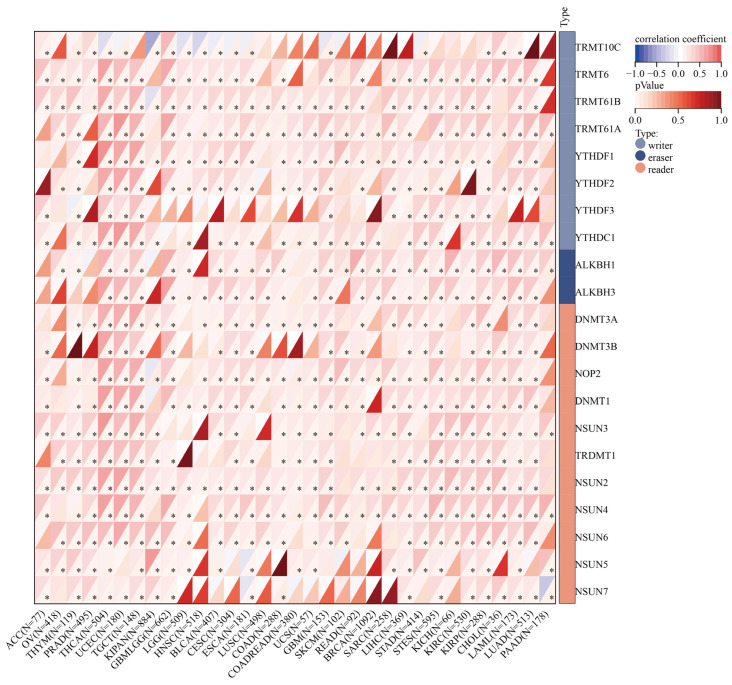
Relationship between SLC2A1 expression and RNA m6A-methylation-related genes in pan-cancer (*, *p* < 0.05).

**Figure 11 cancers-14-05344-f011:**
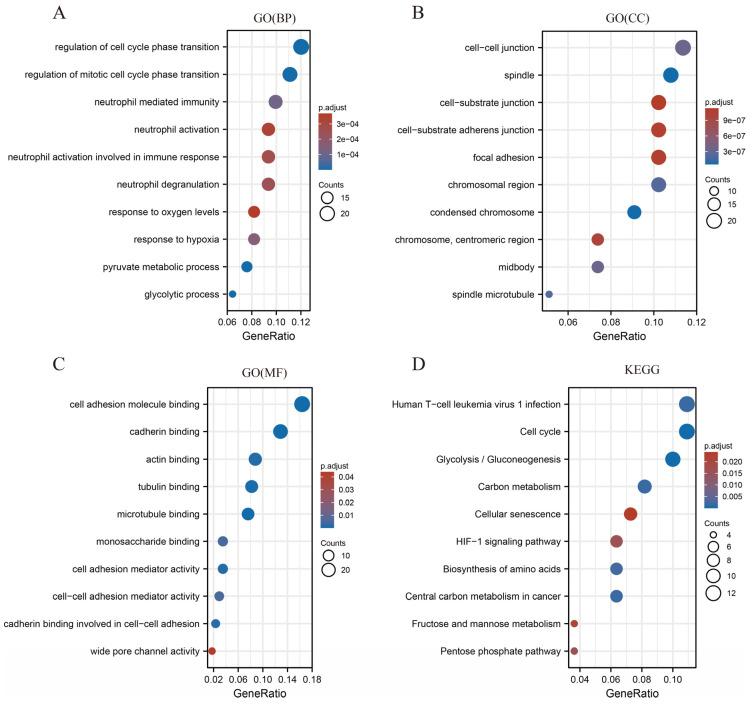
GO (**A**–**C**) and KEGG (**D**) functional enrichment analyses of SLC2A1-related DEGs in TCGA LUAD.

**Figure 12 cancers-14-05344-f012:**
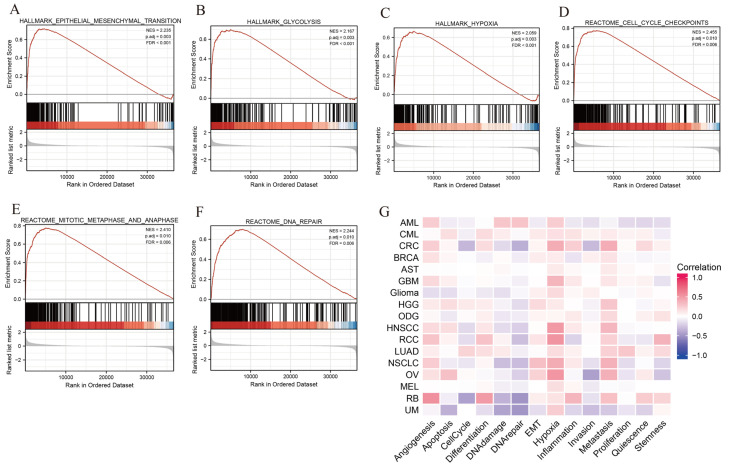
GSEA of SLC2A1-related DEGs based on HALLMARK gene sets (**A**–**C**) and based on REACTOME gene sets (**D**–**F**); single-cell analysis based on the CancerSEA database (**G**).

**Figure 13 cancers-14-05344-f013:**
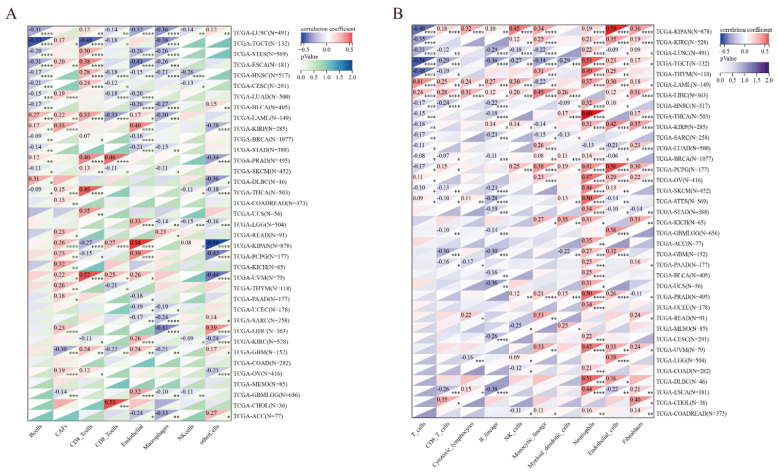
Correlation between SLC2A1 and immune cell infiltration. Heatmap represents correlation between SLC2A1 expression and immune cell infiltration using the EPIC (**A**) and MCPcounter (**B**) algorithms (*, *p* < 0.05; **, *p* < 0.01; ***, *p* < 0.001; ****, *p* < 0.0001).

**Figure 14 cancers-14-05344-f014:**
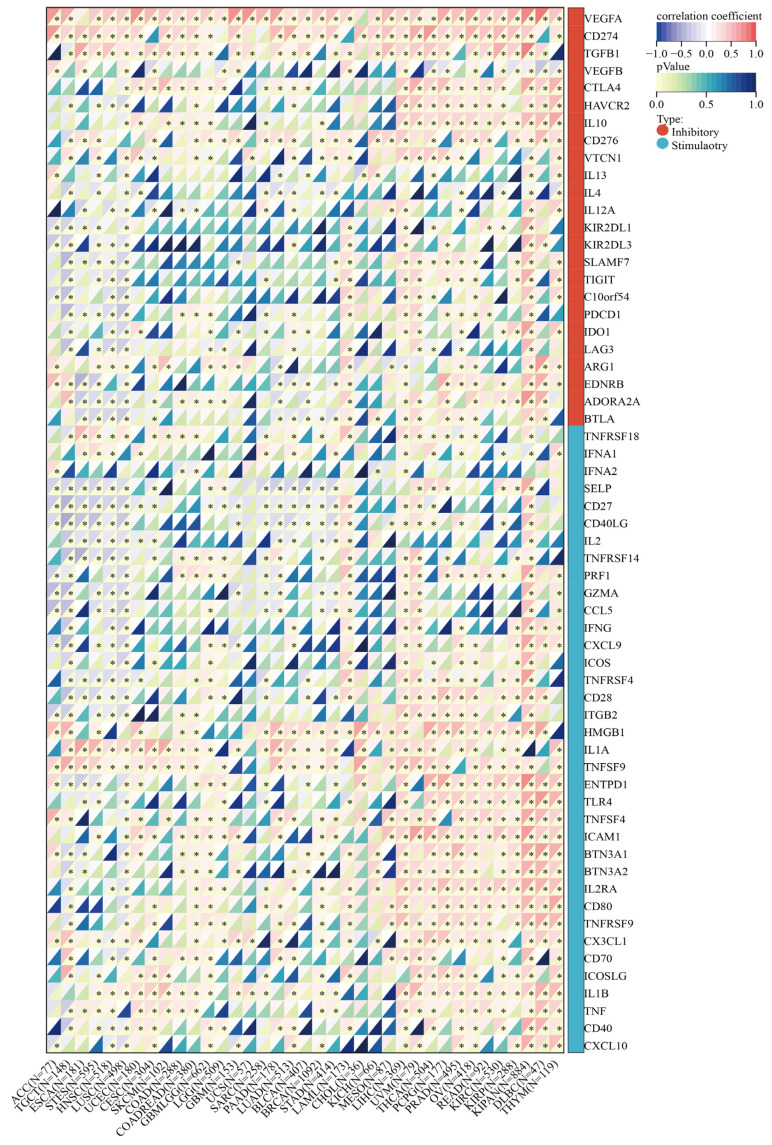
Correlation between SLC2A1 and ICP genes (*, *p* < 0.05).

**Figure 15 cancers-14-05344-f015:**
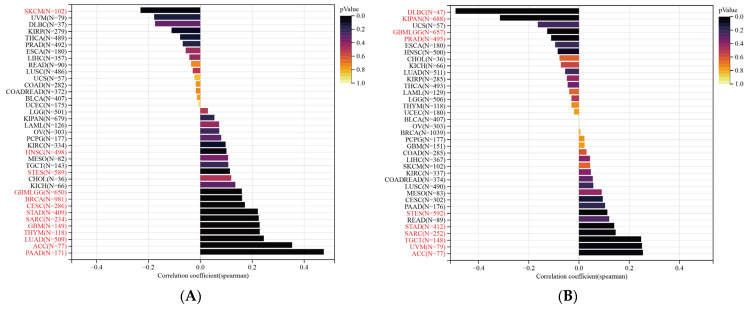
Correlation of SLC2A1 expression with TMB (**A**) and MSI (**B**) in pan-cancer.

## Data Availability

Publicly available datasets were analyzed in this study. TCGA and GTEx data can be found here: (UCSC) Xena browser (https://xena.ucsc.edu/, accessed on 14 July 2022) and the Genotype-Tissue Expression (GTEx) database (https://www.gtexportal.org/home/-index.html, accessed on 14 July 2022); GSE2088, GSE13507, GSE10927, GSE39001, GSE26566, GSE18520, GSE53757, GSE62452, GSE87211, GSE15605, GSE33630, GSE3218, GSE17025, GSE47861, GSE68468, GSE53625, GSE13601, GSE57927, GSE75037, and GSE26899 datasets from GEO database( https://www.ncbi.nlm.nih.gov/geo, accessed on 14 July 2022). The original contributions presented in the study are included in the article. Further inquiries can be directed to the corresponding authors.

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
