# Peer review of "Glycolysis-Related SLC2A1 Is a Potential Pan-Cancer Biomarker for Prognosis and Immunotherapy"

_cancers, 2022, doi:10.3390/cancers14215344_

Round 1

Reviewer 1 Report

In this manuscript, authors explore the expression and function of glucose transporter gene SLC2A1 in pan-cancer. It must be acknowledged that the work of the author is worthy of recognition, but the current version is need major revision to be published. The specific comments are as follows:

1. The authors need to edit the manuscript for a more accurate use of the English language.

2. The Simple Summary section should be a review of the potential role of this study in the field, not a simple repetition of the Abstract section, which needs to be revised and re-conceived.

3. In the results section, the author only gives a simple description of Figure, rather than obtaining directional results. This is not the purpose of Article. In addition, the author should investigate the relevant articles to ensure that the analysis results are consistent with the published literature. If there are inconsistencies, what are the possible reasons? These can also appear in the discussion section.

4. The discussion section should not be a repetition of the above results, but should be a verification of the results of the analysis in conjunction with the results of published literature.

5. There are many problems in the text, such as capitalization, misuse of spaces and abbreviations, which need to be checked and corrected. For example, line 343 DLGAP5.

6. There is no substantial correlation in each of the results sections of this article, and authors should consider linking them in the hope of finding the exact role of SLC2A1 in cancer therapy, rather than generalizing.

7. It is hoped that the author can increase the reading and citation of the literature. Although the bioinformatics analysis lacks the proof of experimental data, the credibility of the analysis results can be increased through the evidence of the published results. As the starting point of glucose uptake, SLC2A1 plays an important role in glucose metabolism and is a target worthy of further study.

Reviewer 2 Report

In this paper, author do a bioinformatics analysis about SLC2A1, a glucose transporter, in Pan-Cancer samples. Indicating SLC2A1 could be a biomarker for prognosis and therapy of cancers. I think this paper can be accepted for publication after authors address the questions below.

Major

1.     Figure 1A, authors should show detail names of every cancer types, but not just abbreviation.

2.     Figure 2U and V, as this paper investigate pan-cancer, authors should validate SLC2A1 expression in more cancer types, at least add one or two cancer types here.

3.     Figure 3C and D, these 2 figures can not support authors’ conclusion, expression seemingly increase in stage 3 and 4 just because expression decrease suddenly in stage 2.

4.     Figure 9, as alteration rate of SLC2A1 in cancers was relatively low. This part is not so important, even I think it is unnecessary. Please remove it or just put it to supplementary data.

5.     Figure 10, for epigenetic analysis, acetylation modification on promoter is also very important. Authors should also analyze acetylation.

Minor

1.     Line 20, authors should show what CAF is here.

Round 2

Reviewer 1 Report

accept!